

# Temporal analysis and opinion dynamics of COVID-19 vaccination tweets using diverse feature engineering techniques

Shoaib Ahmed[1], Dost Muhammad Khan[1], Saima Sadiq[2], Muhammad Umer[1], Faisal Shahzad[1], Khalid Mahmood[3], Heba Mohsen[4] and Imran Ashraf[5]

[1] Department of Computer Science & Information Technology, The Islamia University of Bahawalpur, Bahawalpur, Pakistan
[2] Department of Computer Science, Khwaja Fareed University of Engineering and Information Technology, Rahim Yar Khan, Pakistan
[3] ICT, Gomal University, Dera Ismail Khan, Pakistan
[4] Computer Science Department, Future University in Egypt, New Cairo, Egypt
[5] Information and Communication Engineering, Yeungnam University, Gyeongsan si, South Korea

Corresponding authors
Dost Muhammad Khan,
khan.dostkhan@iub.edu.pk
Imran Ashraf,
imranashraf@ynu.ac.kr

## ABSTRACT

The outbreak of the COVID-19 pandemic has also triggered a tsunami of news, instructions, and precautionary measures related to the disease on social media platforms. Despite the considerable support on social media, a large number of fake propaganda and conspiracies are also circulated. People also reacted to COVID-19 vaccination on social media and expressed their opinions, perceptions, and conceptions. The present research work aims to explore the opinion dynamics of the general public about COVID-19 vaccination to help the administration authorities to devise policies to increase vaccination acceptance. For this purpose, a framework is proposed to perform sentiment analysis of COVID-19 vaccination-related tweets. The influence of term frequency-inverse document frequency, bag of words (BoW), Word2Vec, and combination of TF-IDF and BoW are explored with classifiers including random forest, gradient boosting machine, extra tree classifier (ETC), logistic regression, Naïve Bayes, stochastic gradient descent, multilayer perceptron, convolutional neural network (CNN), bidirectional encoder representations from transformers (BERT), long short-term memory (LSTM), and recurrent neural network (RNN). Results reveal that ETC outperforms using BoW with a 92% of accuracy and is the most suitable approach for sentiment analysis of COVID-19-related tweets. Opinion dynamics show that sentiments in favor of vaccination have increased over time.

# INTRODUCTION

The COVID-19 outbreak changed the lives of people economically and socially. The global pandemic caused life-threatening fears and anxiety in public and many of such concerns have been shared on social media platforms. Social media platforms continuously spread the latest information globally about viruses and provide help to deal with this situation.

As reported in Statistica, 2.9 billion people used social media in 2019 and the number of visitors is expected to exceed 3.4 billion by 2023 (*Dixon (2023)*). Various surveys have been performed by researchers to observe the social media trends which show the high influence of social media platforms for sharing news and stories. *Ofcom (2019)* confirms the wide use of social media for news and updates by adults. Similarly, different departments and disease control institutes such as the Centers for Disease Control and Prevention (CDC) and the World Health Organization (WHO) use social networks for providing the latest updates and guidelines about pandemic emergencies. Quarantine, lockdown, and social distancing constraints intensified the use of social platforms globally (*Hiscott et al., 2020*). Individuals express their emotions and opinions during the different events on this rapidly growing platform (*Alamoodi et al., 2020*). People rely on updates from social media which makes it an influential channel of communication around the globe.

Despite the appropriate and controlled use of social media by WHO and other public institutions, a tsunami of false information has also been produced on social media, creating a big challenge for information systems. Although the United Nations (UN) warned against the COVID-19 infodemic and spread in February 2020 (*Appel et al., 2020*), many malicious users deliberately spread confusion, rumors, and fake news on social media platforms. A famous fake claim that went viral in Europe was that 5G weakens the immune system and is a reason for the spread of COVID-19 and people started demolishing towers (*Schumaker, Jarmoszko & Labedz, 2016*). Many researchers and news reporters highlighted the infodemic issues and discussed case studies to present real information and help people avoid panic. Various posts and ads use the COVID-19 context and mislead the user to install spyware and other cyberattacks. As a result, management and government organizations focus on social media platforms to put a stop to the spread of viral fake news and misinformation. However many platforms are claiming to control this situation by banning harmful content but these platforms were not ready for such information flooding.

Twitter is a famous social media platform where people post their opinion on specific topics in text form called 'tweet' (*D'Andrea et al., 2019*). A tweet also contains the location information of the user, hashtags, and emoticons that help in sentiment portrayal (*Giachanou & Crestani, 2016*). Moreover, Twitter is used by government officials to share information about an event or an announcement for the general public (*Golbeck, Grimes & Rogers, 2010*). Information shared on Twitter has been used in various research works such as analyzing services (*Tiwari et al., 2018*), sports sentiment (*Yu & Wang, 2015*), political views (*Khatua, Khatua & Cambria, 2020*), the sentiment of cancer patients (*Crannell et al., 2016*) and vaccines (*D'Andrea et al., 2019*), *etc.* The subject of vaccination is currently a large debate on social media platforms with respect to questions related to its safety, immunity against the virus, side effects, *etc.* Studies have been conducted to analyze vaccination hesitation and the effect of social media campaigns (*Pedersen et al., 2020*; *Loft et al., 2020*). In general, people show positive and negative opinions on the efficacy of vaccines and the vaccination process itself. Analyzing such opinions from Tweets can help understand the dynamics of vaccination and devise effective policies and social media campaigns to increase vaccination acceptance by the public.

Machine learning approaches have been employed to identify misinformation on social media posts regarding COVID-19. Similarly, public opinions about the COVID-19 vaccination have been studied when two famous vaccines Pfizer and BioNTech were introduced (*Cotfas et al., 2021*). *Mourad et al. (2020)* analyzed 800 k tweets and stated that 93% of tweets are misleading about COVID-19 and from non-medical users and real doctors and medical experts contribute less than 1%. In order to develop effective tactics that might lessen anti-vaccination sentiments among various groups, research that can make use of the large amount of data created *via* social media, such as Twitter, will be able to give important information. To identify trends in vaccination tweets on Twitter, one of the first challenges in this context is to create a text categorization system. The enormous volume of data and text-based style make it a difficult process to complete. Using machine learning techniques was a successful strategy used in various research works conducted on Twitter about vaccination sentiment analysis. Likewise, a large number of Tweets are available on social media platforms which can be used to analyze public opinions about vaccination and devise policies accordingly.

The study aims at investigating the impact of different feature approaches regarding the sentiment classification of COVID-19. Although several existing works investigated and explored similar dimensions, the role of various feature engineering approaches is not very well studied. For this purpose, a dataset containing COVID-19 vaccination-related tweets has been collected and analyzed using machine learning models. In the first instance, the dataset is subdivided into five sub-datasets concerning the administered vaccines AstraZeneca, Moderna, Pfizer/BioNTech, Sinopharm, and SputnickV. Each sub-dataset is investigated separately to analyze people's sentiments and a comparison analysis is performed to discuss the trends. Furthermore, opinion dynamics and temporal analysis are also performed. This study uses a large dataset in this regard and performs sentiment analysis using Tweets on COVID-19 vaccination. This study makes the following contributions:

- A machine learning-based framework was developed for sentiment analysis of tweets related to different vaccines for COVID-19. The sentiments of people for different vaccines were analyzed using several models including random forest (RF), gradient boosting machine (GBM), extra tree classifier (ETC), logistic regression (LR), naive Bayes (NB), stochastic gradient descent (SGD), multilayer perceptron ((MLP), convolutional neural network (CNN), bidirectional encoder representations from transformers (BERT), long short-term memory (LSTM), and recurrent neural network (RNN).
- The influence of term frequency-inverse document frequency (TF-IDF), bag of words (BoW), Word2Vec, and feature union of TF-IDF and BoW was investigated regarding the accuracy of models. Since different feature engineering approaches lead to the different classification accuracy of the models, four feature engineering approaches were investigated regarding high accuracy.

- A large dataset containing Tweets on COVID-19 vaccination was used. For performance comparison, the dataset was labeled manually, as well as using TextBlob. Performance was evaluated in terms of accuracy, precision, recall, and F1-score.

The remainder of the article is arranged as follows. Section 2 discusses the related research work along with their used techniques. Section 3 presents the methods and techniques, dataset, and models used for experiments. It also illustrates the proposed framework. Section 4 provides the experimental results while discussions are provided in Section 5. In the end, Section 6 concludes the article.

## RELATED WORK

Owing to the increase in the data available on social media platforms, there is a need to address various challenges regarding data shape such as information extraction by data restructuring and selection of appropriate classifiers (*Samuel, Kashyap & Betts, 2018*). Text analysis involves text visualization, exploring syntactic and semantic features, and feature extraction techniques (*Samuel, Kashyap & Betts, 2018*; *Rustam et al., 2020*). With the wide use of social media platforms, a large number of opinions and reviews are available on review sites, forums, blogs, *etc.* With the help of review-based prediction systems, this unstructured information can automatically be transformed into structured data of public opinions. This structured data can later be used to find the sentiments about specific applications, products, services, and brands and serves as a piece of important information for product and service refinement.

Twitter data has been widely explored by previous researchers over the years regarding topic modeling, information retrieval, product positioning, and analysis of psychological conditions. Text analysis using tweets has been performed in many types of research such as opinion mining (*Naseem et al., 2019*), aggression detection (*Sadiq et al., 2021*), content mining (*Majumdar & Bose, 2019*), and topic detection related to COVID-19 (*Garcia & Berton, 2021*). Analysis regarding COVID-19 tweets has been performed covering different perspectives such as COVID-19 detection (*Castiglione et al., 2021b*), the role of the internet of things (IoT) to control COVID-19 spread (*Castiglione et al., 2021a*), productivity analysis (*Shoukat et al., 2021*), and effect on mental health (*Sohail et al., 2021*), *etc.* A French company's customer feedback has been analyzed on approximately seventy thousand tweets in *Pépin et al. (2017)*. The authors apply frequency-based feature extraction techniques and topic modeling is done using the Latent Dirichlet Allocation (LDA) method. The authors utilized linguistic and psychological features to explore emotions in social media posts of different languages (*Jain, Kumar & Fernandes, 2017*).

Twitter data has been also used for tracking and analyzing crisis situations during epidemics (*Ye et al., 2016*). Sentiment analysis on Twitter data related to healthcare is carried out regarding the postnatal behavior or depression of new mothers to find their emotions, language style, and social involvement (*De Choudhury, Counts & Horvitz, 2013*). The authors highlighted government policies during the pandemic and performed topic modeling using multi-lingual Twitter data (*Chun et al., 2020*). Similarly, study *Garcia-Gasulla & Suzumura (2020)* analyzed the growth of sinophobia during pandemics

from Twitter data. The study concludes that depression during a pandemic is mainly caused by unemployment, fear of death, and inactive staying at home.

Researchers are exploring tweets from different perspectives using the expressed sentiments toward the COVID-19 pandemic. Tweets from twenty days of March 2020 are collected from Europe and analyzed for the impact of COVID-19 disease spread (*Alhajji et al., 2020*). The authors applied different unsupervised machine-learning models to explore COVID-19-related textual data. Tweets sentiment analysis is done using Naive Bayes with the topic modeling using the LDA in *Prabhakar Kaila & Prasad (2020)*. Similarly, TextBlob and the natural language processing toolkit (NLTK) library are used for the same purpose by *Kaur & Sharma (2020)*. The authors investigated the impact of COVID-19 symptoms on quarantine in *Pastor (2020)*.

Along the same lines, sentiments of the public in China related to COVID-19 are explored by researchers in *Han et al. (2020)*. They divided posts into general seven categories and thirteen subcategories based on topics. *Radwan & Radwan (2020)* discussed that panic caused by COVID-19 by posts on social media is inevitable and spread with more speed than COVID-19 itself. The study further states that public behavior, sentiments, and rumors need to be investigated quickly by experts to assist authorities in taking action accordingly. Similarly, the study analyzed the emotions of the general public using the data from the discussion forum to conclude that Twitter posts have the highest influence on the behavior of people (*Hanson et al., 2013*).

Lexicon-based, machine learning, and hybrid techniques are mostly used by researchers for polarity analysis. Lexicon techniques include sentiment lexicons like SentiWordnet (*Baccianella, Esuli & Sebastiani, 2010*), VaderSentiment (*Hutto & Gilbert, 2014*), and sentiment140 (*Mohammad, Kiritchenko & Zhu, 2013*) consisting of words and polarity score. The sentiment lexicons are utilized with semantic approaches, which commonly take negations and booster words into account, to accomplish polarity identification. *Hutto & Gilbert (2014)* proposed Vader, a simple rule-based model that incorporates a sentiment lexicon as well as syntactic and grammatical rules. The authors demonstrate that the suggested model performs better than a single human rater. The authors demonstrate that Vader gives a higher performance on the datasets gathered from Twitter, Amazon reviews, and NYT editorials when compared to traditional machine learning models.

In order to track the dynamics of emotions in the first few months after the public learned about COVID-19, *Kaur, Kaul & Zadeh (2020)* utilized data taken from Twitter. The IBM Watson Tome Analyzer was used to extract and analyze a total of 16,138 tweets. In all three months analyzed in the article, more negative tweets were sent than neutral or good ones, as was to be expected. Deep learning and transfer learning models have been employed for cross-domain sentiment analysis. Deep learning models' capability of transferability improves the performance and avoids overfitting (*Cao et al., 2021*). The adversarial training model is proposed in *Dai et al. (2022)* to transfer sentiments across domains. Authors applied decision boundaries in cross-domain sentiment analysis (*Fu & Liu, 2022*). Authors mine keywords and applied feature engineering techniques to explore patterns (*Asgarnezhad, Monadjemi & Aghaei, 2022*). Aspect-level sentiment analysis has

been performed using an adaptive SVM model and Twitter dataset (*Liu et al., 2022*). *Du et al. (2022)* applied a gated attention model for sentiment classification.

The opinions of the general population on 11 chosen topics using Latent Dirichlet Allocation were examined by *Xue et al. (2020)* utilizing data from COVID-19 tweets. The findings are consistent with earlier research on COVID-19 that claims that the coronavirus epidemic has a major influence on people's psychological states, according to the authors, who also determined that fear is the most prevalent emotion across all of the themes they investigated.

Despite the above-mentioned studies, the sentiments related to different vaccines and vaccination acceptability is an under-investigated area and requires further research. This study presents a detailed analysis in this regard by obtaining the data for different vaccines and a separate analysis for each vaccine.

## MATERIALS AND METHODS

This section discusses the dataset, feature engineering techniques, machine learning, deep learning models, and proposed methodology used to analyze the COVID-19 vaccine-related sentiments on manually labeled Twitter data.

### Dataset description

This study uses the dataset, 'COVID-19 All Vaccines Tweets', which was obtained from the Kaggle repository (*Preda, 2022*). The dataset contains tweets related to COVID-19 vaccines. Twitter data are used for the following reasons:

- For social media platforms like Facebook, users need to be friends with each other before they can follow others because it is based on friendship pattern (*Stieglitz & Dang-Xuan, 2013*). In contrast, there are no such restrictions on Twitter; anyone can follow others according to their interest.
- Soft policy of Twitter for developers to access their data. The 'no friendship' pattern of Twitter makes it more vulnerable to spreading misinformation rapidly. Such platforms need more attention to using automatic detection methods to moderate their discussions.

Tweets were related to five vaccines including 'AstraZeneca', 'Moderna', 'Pfizer/ BioNTech', 'Sinopharm' and 'Sputnik V'. The dataset was labeled manually. Tweets classified into the former class present in favor opinions of users, while tweets under the latter class present negative comments the users regarding COVID-19 vaccination. The dataset was divided into two classes: 'Against' and 'In favor'. A few sample reviews are given in Table 1. Table 2 shows the distribution of labels.

### Feature engineering techniques

Feature engineering techniques are used to extract appropriate information from raw data to train machine learning models (*Bocca & Rodrigues, 2016*). The feature engineering process is required for machine learning models and their performance is affected by the choice of the feature engineering method (*Heaton, 2016*). This process converts the data

**Table 1 Sample reviews for tweets related to COVID-19 vaccination.**

| Vaccine | Tweet | Sentiment |
|---|---|---|
| Sinopharm | The vaccine manufacturers have said that their formulas are effective against the new variant. | In favor |
| | Feeling pain in my shoulder after getting first dose of vaccine, not recommended | Against |
| AstraZeneca | Do not take the vaccine. | Against |
| | Good morning. I had my COVID vaccination yesterday—feeling fine! | In favor |
| SputnikV | Good grief. This is just pure evil. | Against |
| | #SputnikV #COVID-19 Russian vaccine is created to last 24 years effective | In favor |
| PfizerBioNTech | #PfizerBioNTech COVID vaccine is not safe whilst breastfeeding | Against |
| | COVID vaccine you getting it #COVIDVaccine #Pfizer/BioNTech | In favor |
| Moderna | While the world has been on the wrong side of history this year, hopefully the biggest vaccination effort ever. | Against |
| | There have not been many bright days in 2020, but here are some of the best #Moderna. | In favor |

**Table 2 COVID-19 vaccination tweets labeling using VADER and TextBlob.**

| Vaccine name | VADER | | TextBlob | | Manual labeling | |
|---|---|---|---|---|---|---|
| | In favor | Against | In favor | Against | In favor | Against |
| AstraZeneca | 1,393 | 1,980 | 1,408 | 1,965 | 1,768 | 1,605 |
| Moderna | 9,206 | 13,178 | 9,209 | 13,175 | 13,739 | 8,645 |
| PfizerBioNTech | 2,038 | 2,906 | 2,052 | 2,892 | 2,975 | 1,969 |
| Sinopharm | 2,053 | 2,519 | 2,084 | 2,488 | 2,557 | 2,015 |
| SputnikV | 2,977 | 5,526 | 2,972 | 5,531 | 5,718 | 2,785 |

into a feature vector, suitable to train the models. In this work, three feature engineering techniques BoW, TF-IDF, Word2vec, and feature union (TF-IDF+BoW) are used. The advantages and disadvantages of feature engineering techniques are presented in Table 3.

### Bag of words

The BoW is a simple and widely used technique to extract features from raw text. It is easy to implement and is mostly used in text categorization and language modeling. It uses CountVectorizer for feature extraction by considering term occurrences in the form of a matrix (*Eshan & Hasan, 2017*). Each feature or word in a matrix is assigned a value according to its number of occurrences in the *corpus* (*Hu, Downie & Ehmann, 2009*).

### Term frequency-inverse document frequency

TF-IDF is another commonly used technique for feature extraction from raw text data. It is mostly used in textual information retrieval and text classification (*Yu, 2008*). In contrast to simple term count in BoW, TF-IDF also assigns weights to each word regarding its importance. It was done using inverse document frequency along with term frequency (*Robertson, 2004*). Important terms are represented with higher weight values. It can be calculated using

**Table 3 Advantages and disadvantages of feature representation technique.**

| Technique | Type | Advantages | Disadvantages |
|---|---|---|---|
| TF | Vectorization technique | –Calculate the frequency of a document's most frequently used term.<br>–Count the number of times each word appears. | –The issue with using raw word frequency data is that adding relevance does not make usage more proportionate. |
| TF-IDF | Vectorization technique | –Quickly compare documents for similarities.<br>–Calculate the frequency of each distinct term in a text as well as the entire *corpus*.<br>–Weight is inversely correlated with word frequency within texts and directly correlated with word frequency inside documents.<br>–Stop words like is, a, *etc*. have less impact than uncommon words. | –Enormous vector size.<br>–Position and its co-occurring phrases are not taken into consideration.<br>–Do not take semantics and context into account.<br>–Sparsity problem.<br>–It is ineffective to distinguish polysemy terms and compare similarities between synonyms. |
| BoW | Vectorization technique | –Simple and easy to use-offers feature representation of free-form text for NLP tasks.<br>–Words to vectors mapping | –Large vocabulary makes it challenging to train the model.<br>–Sparsity matrix.<br>–Our vocabulary would expand if the new phrases included new terms, which would also lengthen the vectors. |
| Word2Vec | Prediction based technique | –Works on words' probability.<br>–Map words to target vectors.<br>–CBOW predicts the words' probability and skip-gram determines the words' context. | –Large-sized vocabulary make the model difficult to train on Word2Vec.<br>–Consider word similarities.<br>–CBOW Take polysemy words' average, separate vectors are used to present skip-gram. |

$$W_{i,j} = [1 + log(tf_{i,j})] \times \left[ log\left( \frac{N}{df_i} \right) \right] \qquad (1)$$

where $N$ in the total number of documents, $TF_{i,j}$ represents term frequency in document and $D_{f,t}$ is the document containing term $t$.

### Word2Vec

The Word2Vec model extracts the idea of similarity between words or items, such as semantic similarity, synonym identification, concept classification, selectional preferences, and analogies. In word embedding, words that have the same meaning are represented similarly, which is a learned representation for text (*Egger, 2022*). One of the major advances in deep learning for difficult natural language processing tasks may be attributed to this method of encoding words and documents. Word embeddings are n-dimensional distributed representations of text. These are necessary for resolving the majority of NLP issues.

### Feature union

The methodology used for feature union is presented in Fig. 1. Features are extracted using TF-IDF and BoW separately and are joined to enlarge the feature vector.
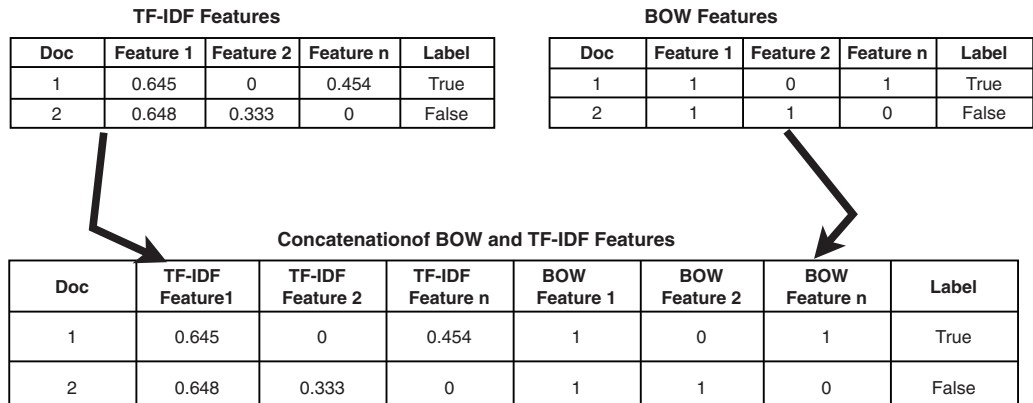

**Figure 1   Methodology adopted for feature union.**

## Models

This research takes advantage of various machine learning models (such as ensemble learning classifiers, regression-based models, and probability-based models), and deep learning models to classify tweets related to COVID-19. This study considers the use of the following classifiers for sentiment analysis of COVID-19 vaccine-related tweets as presented in Table 4. All classifiers are implemented using Sci-kit library (*Pedregosa et al., 2011*).

## Proposed methodology

This section discusses the proposed methodology to investigate the COVID-19 vaccine-related sentiments from Twitter data.

Figure 2 shows the architecture of the proposed framework. At first, the Twitter dataset related to COVID-19 is divided into five subsets according to vaccine types AstraZeneca, Moderna, Pfizer/BioNTech, Sinopharm, and SputnickV. The dataset contains the highest number of tweets regarding these vaccines. Each subset is analyzed individually and each tweet is classified as 'In Favor' or 'Against'. Data goes through preprocessing steps like stopwords removal, number removal, special character removal, lemmatization, and tokenization. The dataset is labeled with the help of graduate students from the artificial intelligence department. Each vaccine subset is assigned to three students. For labeling, the following criteria are used:

- Three students label the data separately,
- A label is assigned, if at least two of the annotators agree,
- In case of a different label for a tweet from each annotator, the tweet is dropped.

After labeling, datasets are prepared to train the machine learning models. Dataset is split into training and testing in the ratio of 70% and 30%, respectively. Then feature engineering techniques are applied to both training and test sets. This study uses BoW, TF-IDF and their union (BoW+TF-IDF) with supervised machine learning models to

**Table 4 Description of machine learning and deep learning models.**

| Reference | Model | Description |
|---|---|---|
| Breiman (2001) | RF | RF is one of the meta-estimators that integrate aggregation of a number of decision trees (DT) in order to provide improved efficacy and reduced over-fitting of the framework. It works by fitting DT classifiers on a number of samples of the input data. Afterward, it averages the results obtained from each DT classifier thus working as an ensemble learner. |
| Friedman (2001) | GBM | GBM is an ensemble model that develops an additive model in an optimized manner by the integration of a loss function. It works in an iterative manner that optimizes the error rate at each iteration by using the loss function. The purpose of the gradient boosting algorithm is to specify the outcomes of the target variable for the next model to lessen the prediction error. |
| Sharaff & Gupta (2019) | ETC | ETC works similarly to the RF model and a tree-based model. It is also known as an extremely randomized tree and unlike RF, it does not use bootstrap data, it builds trees from the actual data samples. It was proposed to build trees by considering the numeric input and selecting optimal cut-point to avoid variance at each node which reduces the computational complexity. |
| Boyd, Tolson & Copes (1987) | LR | LR works on a probability-based model and is used for classification tasks. It uses a logistic function for the modeling of binary variables. LR utilizes the correlation coefficient which is the measure of the relationship between the target variable and the independent variable. |
| Pérez, Larrañaga & Inza (2006) | NB | NB is based on 'Bayes' theorem which works on the assumption of independent features. It focuses on the prior probability and posterior probability of a target label in the dataset. Its supposition of considering feature independence is unrealistic for actual data. It shows robust results on large-sized and complex data having multiple classes. |
| Gardner (1984) | SGD | SGDC works on a one-versus-all technique. It is an optimization algorithm and finds the best suitable features or parameters among predicted and actual target values (Gardner, 1984). It gives good results on the large-sized dataset and uses a maximum sample at each iteration. It is sensitive regarding hyperparameter tuning. |
| Kocyigit, Alkan & Erol (2008) | MLP | MLP has significant characteristics with respect to classification such as it is easy and simple to implement. MLP performs well on the small-sized training set. MLP consists of mainly three layers that are hidden layers, the input layer, and the output layer. |
| Krizhevsky, Sutskever & Hinton (2012) | CNN | CNN is a deep neural network and efficiently learns features with the help of pooling layers, non-linear activation, dropout layer, and most importantly convolution layers. It was first developed for image-based tasks such as image categorization and image segmentation. End-to-end training makes CNN more efficient. |
| Škrlj et al. (2019) | RNN | Recurrent neural networks are a type of artificial neural network in which connections between nodes can form a cycle, allowing the output of certain nodes to influence input to other nodes in the same network in the future. This enables it to display temporal dynamic behavior. |
| Staudemeyer & Morris (2019) | LSTM | An artificial neural network called long short-term memory is utilized in deep learning and artificial intelligence. LSTM features feedback connections as opposed to typical feedforward neural networks. LSTM may analyze complete data sequences in addition to single data points. |
| Yang & Cui (2021) | BERT | Google has created a transformer-based machine learning method for pre-training natural language processing called Bidirectional Encoder Representations from Transformers (BERT). It is an open-source model and pre-trained on a large volume of data and often performs well. |

select features from COVID-19 vaccine-related tweets. Models are optimized using several hyperparameters which are fined tuned, as shown in Table 5.

Machine learning models are trained using three settings of feature extraction methods and then test data is used for performance evaluation in terms of accuracy, precision, recall, and F1-score. The following equations are used for performance evaluation metrics.

$$Accuracy = \frac{TP + TN}{TP + TN + FP + FN} \tag{2}$$

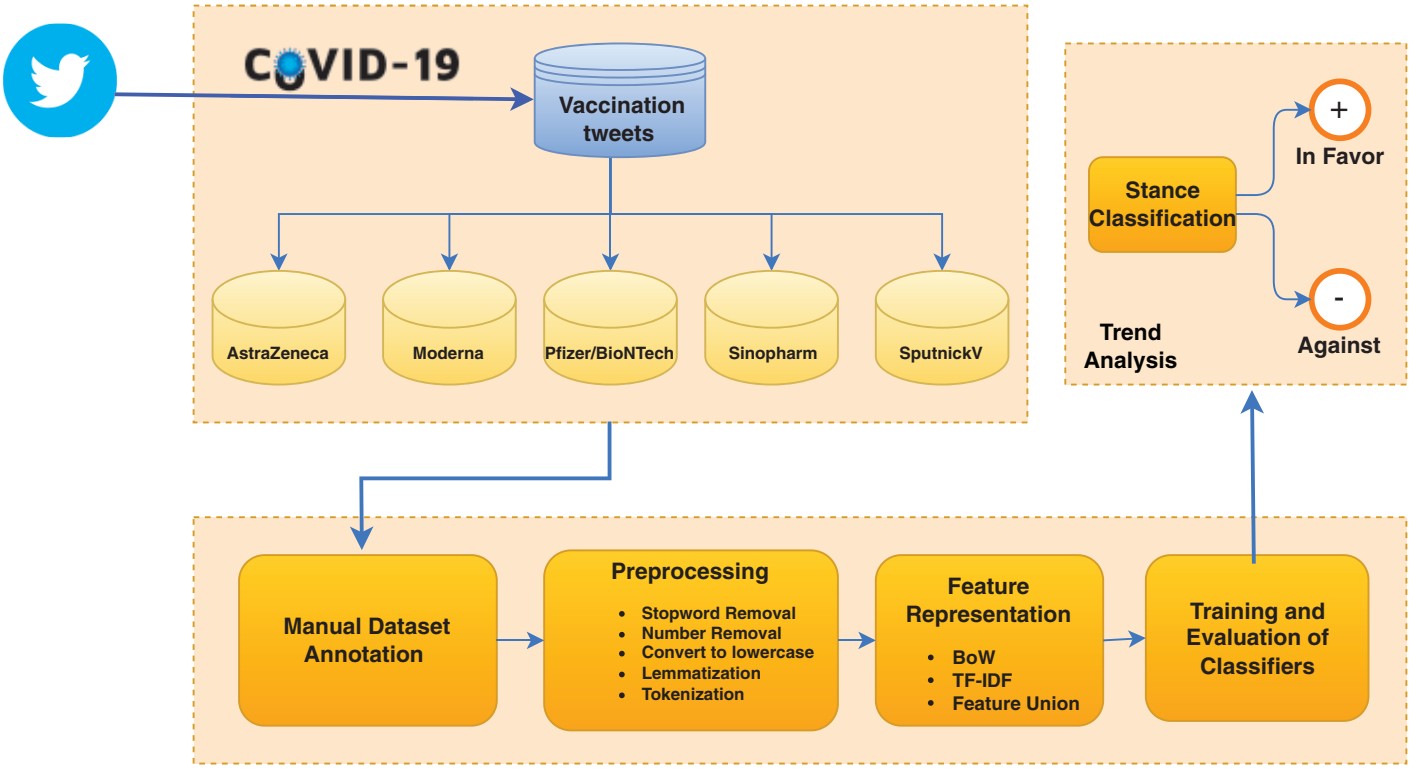

**Figure 2** The architecture of the proposed framework.

**Table 5** Hyperparameter setting of learning models.

| Classifiers | Parameters |
| --- | --- |
| RF | n_estimator=200, max_depth=30, random_state=52 |
| ETC | n_estimator=200, max_depth=30, random_state=52 |
| GBM | n_estimator=200, max_depth=30, random_state=52, learning_rate=0.1 |
| LR | penalty='l2', solver='lbfgs' |
| NB | alpha=1.0, binarize=0.0 |
| SGD | penalty='l2', loss='log' |
| MLP | Dense (neurons=300), dense (neurons=200), dense (neurons=100), activation='relu', dropout (0.5), optimizer='adam', softmax(2) |
| CNN | Conv (7, @64), Max pooling (2×2), Conv (7, @64), GlobalMax pooling (2×2), Dropout (0.5), Dense (32 neurons), optimizer='adam', Softmax (2) |

$$Precision = \frac{TP}{TP + FP} \tag{3}$$

$$Recall = \frac{TP}{TP + FN} \tag{4}$$

$$F1-score = 2 \times \frac{precision \times recall}{precision + recall} \tag{5}$$

where TP, TN, FP, and FN are true positive, true negative, false positive, and false negative, respectively, and extracted from the confusion matrix.

## RESULTS

This study compared the result of machine learning models by combining each model with feature extraction techniques and their union. Machine learning models are implemented using scikit-learn (*Hackeling, 2017*) in Python. Experiments have been performed in different settings and the best parameters are selected following the grid search approach. Algorithms have been evaluated by applying TF-IDF, Word2Vec, BoW, and Feature Union (TF-IDF+ BoW) techniques.

### Experimental results of machine learning models

A set of experiments are performed using the TF-IDF, Word2vec, Feature Union, and BoW features with selected machine-learning models on all five divisions of the dataset. Different features are used on the manually labeled dataset and accuracy results are presented in Table 6. Results reveal that the best-performing model is ETC using BoW on all sub-datasets. ETC using BoW achieved 92% accuracy for the Sinopharm sub-dataset. SGD has also shown good results and ranked second on this dataset using BoW for sentiment analysis of tweets. In the case of the Moderna dataset, SGD using BoW has shown the highest result in terms of accuracy value with 90% which is equal to the value achieved by ETC. The results show that ETC outperforms other models using BoW on all sub-datasets for sentiment analysis.

Class-wise results achieved on the manually labeled dataset are presented in Table 7 separately using the BoW feature with which machine learning models have shown the highest accuracy. It can be observed that ETC outperformed other models with 86% accuracy using BoW on the AstraZeneca dataset. ETC has also achieved the highest scores for precision and F1-score for both the 'In favor' and 'Against' classes. The highest recall is achieved by GBM and LR for 'Against' class each with 97% and by ETC for In favor class with 72%. In the case of the Moderna dataset, the highest accuracy result has been achieved by ETC and SGD with 90% accuracy each. The highest precision for the 'Against' class is 91% by ETC and SGD. For the 'In favor' class, the highest precision is 91% by both NB and LR. The highest recall is 97% for the 'Against' class by NB.

Similarly, results of classifiers with the Pfizer dataset reveal that ETC surpassed other models with 90% accuracy and 92% F1-score for the positive class and 87% F1-score for the negative class using BoW. For the Sinopharm dataset, results show that the highest results in terms of accuracy and precision, and F1-score for the positive and negative classes are achieved by ETC. SGD obtains the second-best results using TF-IDF for sentiment analysis on the Pfizer dataset. The lowest results have been achieved by NB. Subsequently, it can be seen that ETC using BoW has shown the highest results in terms of accuracy and F1-score on the SputnikV dataset. The highest precision for the negative class is 92% achieved by ETC. The highest recall for the negative class is 98% and it is achieved by NB.

**Table 6 Accuracy of models with different features using manually labeled dataset.**

| Dataset | Model | TF-IDF | Word2Vec | Feature union | BoW |
|---|---|---|---|---|---|
| AstraZeneca | RF | 83% | 80% | 81% | 85% |
| | ETC | 84% | 79% | 82% | 86% |
| | GBM | 79% | 70% | 74% | 82% |
| | LR | 84% | 77% | 81% | 83% |
| | NB | 78% | 74% | 79% | 78% |
| | SGD | 84% | 78% | 81% | 85% |
| Moderna | RF | 87% | 83% | 85% | 88% |
| | ETC | 89% | 85% | 88% | 90% |
| | GBM | 85% | 82% | 83% | 84% |
| | LR | 88% | 84% | 89% | 89% |
| | NB | 84% | 80% | 81% | 81% |
| | SGD | 88% | 84% | 89% | 90% |
| Pfizer | RF | 86% | 84% | 86% | 88% |
| | ETC | 88% | 86% | 85% | 90% |
| | GBM | 86% | 87% | 86% | 86% |
| | LR | 87% | 85% | 88% | 88% |
| | NB | 85% | 87% | 86% | 83% |
| | SGD | 87% | 86% | 88% | 89% |
| Sinopharm | RF | 90% | 89% | 88% | 90% |
| | ETC | 89% | 86% | 88% | 92% |
| | GBM | 84% | 85% | 84% | 90% |
| | LR | 87% | 86% | 88% | 89% |
| | NB | 84% | 84% | 86% | 81% |
| | SGD | 87% | 84% | 89% | 91% |
| SputnikV | RF | 87% | 89% | 84% | 90% |
| | ETC | 89% | 87% | 88% | 90% |
| | GBM | 86% | 87% | 86% | 87% |
| | LR | 89% | 81% | 87% | 88% |
| | NB | 83% | 78% | 85% | 81% |
| | SGD | 89% | 83% | 90% | 89% |

## Experimental results of deep learning models

For a fair comparison, experiments are also performed using deep learning models and results are presented in Table 8. Two feed-forward deep learning models including convolutional neural network (CNN) (*Kamath, Liu & Whitaker, 2019*) and multilayer perceptron (MLP) (*Tang, Deng & Huang, 2015*) and three RNN-based deep learning models including bidirectional encoder representations from Transformers (BERT) (*Yang & Cui, 2021*), LSTM (*Staudemeyer & Morris, 2019*), and recurrent neural network (RNN) (*Škrlj et al., 2019*) are used in the experiments. Experimental results reveal that deep learning models using word2Vec have not shown better results in comparison with machine learning models using TF-IDF, BoW, and feature union.

**Table 7 Experimental results of the manually labeled dataset using BoW features.**

| Dataset | Model | Class | Prec. | Recall | F1-score | Accuracy |
|---|---|---|---|---|---|---|
| AstraZeneca | RF | Against | 82% | 96% | 88% | 85% |
| | | In favor | 92% | 66% | 77% | |
| | ETC | Against | 84% | 95% | 89% | 86% |
| | | In favor | 90% | 72% | 80% | |
| | GBM | Against | 79% | 97% | 87% | 82% |
| | | In favor | 93% | 58% | 72% | |
| | LR | Against | 80% | 96% | 87% | 83% |
| | | In favor | 91% | 63% | 74% | |
| | NB | Against | 75% | 95% | 84% | 78% |
| | | In favor | 86% | 50% | 64% | |
| | SGD | Against | 83% | 95% | 88% | 85% |
| | | In favor | 89% | 69% | 78% | |
| Moderna | RF | Against | 88% | 94% | 91% | 88% |
| | | In favor | 89% | 79% | 84% | |
| | ETC | Against | 91% | 93% | 92% | 90% |
| | | In favor | 89% | 86% | 87% | |
| | GBM | Against | 81% | 96% | 88% | 84% |
| | | In favor | 90% | 64% | 75% | |
| | LR | Against | 89% | 95% | 92% | 89% |
| | | In favor | 91% | 81% | 86% | |
| | NB | Against | 78% | 97% | 86% | 81% |
| | | In favor | 91% | 57% | 70% | |
| | SGD | Against | 91% | 94% | 92% | 90% |
| | | In favor | 90% | 85% | 87% | |
| Pfizer | RF | Against | 86% | 95% | 90% | 88% |
| | | In favor | 91% | 76% | 83% | |
| | ETC | Against | 91% | 93% | 92% | 90% |
| | | In favor | 88% | 86% | 87% | |
| | GBM | Against | 83% | 96% | 89% | 86% |
| | | In favor | 93% | 69% | 79% | |
| | LR | Against | 87% | 96% | 91% | 88% |
| | | In favor | 92% | 77% | 84% | |
| | NB | Against | 80% | 96% | 87% | 83% |
| | | In favor | 91% | 63% | 75% | |
| | SGD | Against | 89% | 94% | 91% | 89% |
| | | In favor | 90% | 82% | 85% | |
| Sinopharm | RF | Negative | 87% | 98% | 92% | 90% |
| | | In favor | 96% | 81% | 88% | |
| | ETC | Against | 89% | 97% | 93% | 92% |
| | | In favor | 96% | 85% | 90% | |
| | GBM | Against | 87% | 97% | 92% | 90% |

| Table 7 (continued) | | | | | | |
|---|---|---|---|---|---|---|
| Dataset | Model | Class | Prec. | Recall | F1-score | Accuracy |
| | | In favor | 96% | 81% | 88% | |
| | LR | Against | 87% | 97% | 91% | 89% |
| | | In favor | 95% | 81% | 88% | |
| | NB | Against | 756% | 98% | 85% | 81% |
| | | In favor | 95% | 61% | 74 % | |
| | SGD | Against | 88% | 96% | 92 % | 91% |
| | | In favor | 94% | 85% | 89 % | |
| SputnikV | RF | Against | 90% | 96% | 93% | 90% |
| | | In favor | 90% | 78% | 83% | |
| | ETC | Against | 92% | 95% | 93% | 90% |
| | | In favor | 88% | 81% | 84% | |
| | GBM | Against | 86% | 97% | 91% | 87% |
| | | In favor | 91% | 65% | 76% | |
| | LR | Against | 88% | 96% | 92 % | 88% |
| | | In favor | 89% | 72% | 79% | |
| | NB | Against | 79% | 98% | 87 % | 81% |
| | | In favor | 89% | 44% | 59% | |
| | SGD | Against | 91% | 93% | 92% | 89% |
| | | In favor | 85% | 79% | 82% | |

Results indicate that the best performance of the BERT model is on the Sinopharm dataset where its accuracy is 85%. On the other hand, the LSTM model shows the best performance on the Sinopharm dataset with an 83% accuracy while the precision, recall, and F1-score each is 82%, 80%, and 81% respectively. RNN performed the best using the tweets regarding the Sinopharm vaccination and achieve an accuracy of 80%, precision of 85%, recall of 82%, and F1-score of 83%. Results from these models show lower accuracy as compared to the proposed approach.

## Results of K-fold cross-validation

This study validates the proposed approach by performing a 10-fold cross-validation. Results of 10-fold cross-validation for the proposed model are given in Table 9. Experiments results are provided with respect to each of the five vaccines considered in this study. Results demonstrate the proposed approach shows better results with cross-validation as well.

## Experimental results using COVID-19 VAERS dataset

To prove the effectiveness of the proposed approach, we performed additional experiments using another manually annotated dataset. For this purpose, the COVID-19 VAERS dataset is used which is publicly available on Kaggle. It is a benchmark dataset that contains adverse events reported after COVID-19 vaccination (Garg, 2021). It has a total of 5,351 event reports. This study utilized the multi-class classification problem using the

**Table 8 Result of deep learning classifiers using Word2Vec features.**

| Dataset | Model | Precision | Recall | F1-score | Accuracy |
|---|---|---|---|---|---|
| AstraZeneca | CNN | 78% | 76% | 77% | 80% |
| | MLP | 81% | 85% | 83% | 81% |
| | BERT | 80% | 82% | 81% | 79% |
| | LSTM | 79% | 82% | 81% | 78% |
| | RNN | 76% | 79% | 78% | 74% |
| Moderna | CNN | 80% | 88% | 86% | 82% |
| | MLP | 82% | 89% | 85% | 81% |
| | BERT | 81% | 85% | 83% | 84% |
| | LSTM | 80% | 82% | 81% | 81% |
| | RNN | 77% | 81% | 79% | 80% |
| Pfizer | CNN | 84% | 82% | 83% | 79% |
| | MLP | 81% | 84% | 82% | 81% |
| | BERT | 84% | 80% | 82% | 83% |
| | LSTM | 77% | 79% | 78% | 81% |
| | RNN | 80% | 78% | 79% | 80% |
| Sinopharm | CNN | 84% | 79% | 83% | 87% |
| | MLP | 80% | 80% | 80% | 82% |
| | BERT | 87% | 85% | 86% | 85% |
| | LSTM | 82% | 80% | 81% | 83% |
| | RNN | 85% | 82% | 83% | 80% |
| SputnikV | CNN | 77% | 84% | 81% | 88% |
| | MLP | 76% | 79% | 78% | 85% |
| | BERT | 84% | 84% | 84% | 83% |
| | LSTM | 81% | 79% | 80% | 80% |
| | RNN | 80% | 77% | 79% | 80% |

**Table 9 Results of 10-fold cross-validation on all datasets with the best performing model ETC and BoW features.**

| Fold number | Astrazeneca | Moderna | Pfizer | Sinopharm | SputnikV |
|---|---|---|---|---|---|
| 1st-fold | 0.913 | 0.922 | 0.911 | 0.902 | 0.902 |
| 2nd-fold | 0.898 | 0.913 | 0.909 | 0.911 | 0.923 |
| 3rd-fold | 0.869 | 0.911 | 0.914 | 0.923 | 0.914 |
| 4th-fold | 0.863 | 0.901 | 0.913 | 0.937 | 0.931 |
| 5th-fold | 0.854 | 0.933 | 0.912 | 0.922 | 0.911 |
| 6th-fold | 0.852 | 0.913 | 0.923 | 0.912 | 0.878 |
| 7th-fold | 0.858 | 0.921 | 0.866 | 0.871 | 0.882 |
| 8th-fold | 0.899 | 0.908 | 0.912 | 0.909 | 0.923 |
| 9th-fold | 0.889 | 0.919 | 0.882 | 0.902 | 0.927 |
| 10th-fold | 0.891 | 0.942 | 0.892 | 0.911 | 0.933 |
| Average | 0.8786 | 0.9183 | 0.9034 | 0.9100 | 0.9124 |

**Table 10 Results of machine learning models using BoW on the COVID-19 VAERS dataset.**

| Model | Accuracy | Precision | Recall | F1-score |
|---|---|---|---|---|
| RF | 0.9681 | 0.94 | 0.97 | 0.95 |
| ETC | 0.9701 | 0.94 | 0.97 | 0.96 |
| GBM | 0.9581 | 0.94 | 0.96 | 0.95 |
| LR | 0.9601 | 0.94 | 0.97 | 0.96 |
| NB | 0.9681 | 0.95 | 0.97 | 0.96 |
| SGD | 0.9701 | 0.94 | 0.97 | 0.96 |

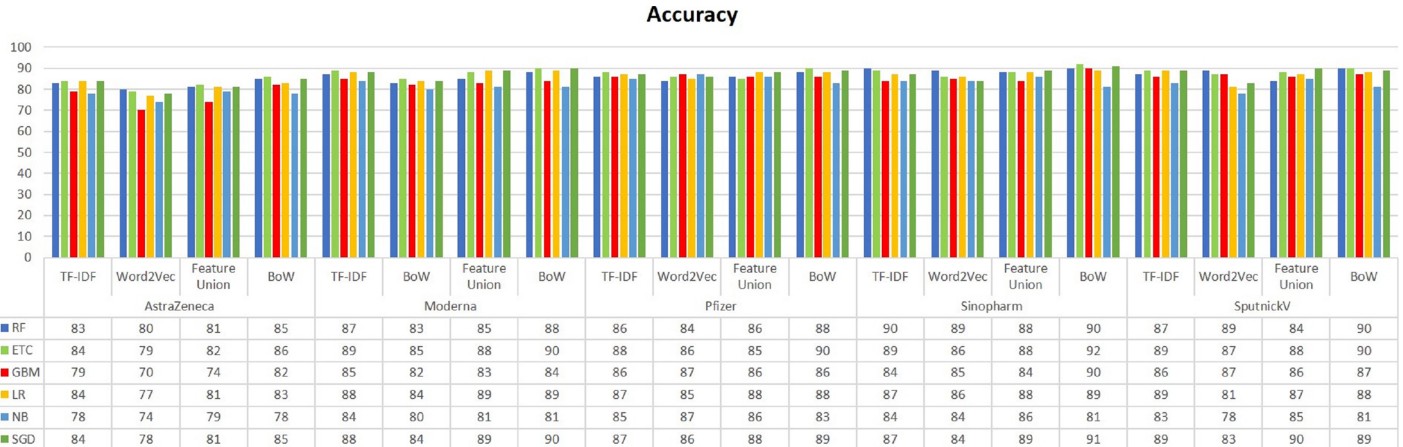

**Figure 3 Accuracy result comparison of machine learning models.**

'recovered', 'not covered', and 'recovery status unknown' classes of the dataset. For complete details of the dataset, the readers are referred to *Saad et al. (2022)*. We implemented the proposed approach on this dataset and the results of multiclass classification are presented in Table 10. Results indicate that the proposed approach shows superior performance on the manually labeled dataset as well. Of the used models, ETC and SGD show the best performance with a 0.97 accuracy score while the precision, recall, and F1-scores are 0.94, 0.97, and 0.96, respectively. In addition, the performance of other models is marginally different.

## DISCUSSION

Performance comparison of classifiers using different feature representation techniques has been carried out on five subsets of datasets based on the COVID-19 vaccine type. The impact of TF-IDF, BoW, and their union (TF-IDF+BoW) has been investigated in tweets to determine the trend of public opinion about COVID-19 vaccines. Comparative analysis in terms of accuracy, precision, recall, and F1-score has been presented separately.

Figure 3 presents the accuracy comparison of classifiers on all five datasets. It can be noted that RF, GBM, ETC, LR, and SGD have achieved the highest accuracy score on every

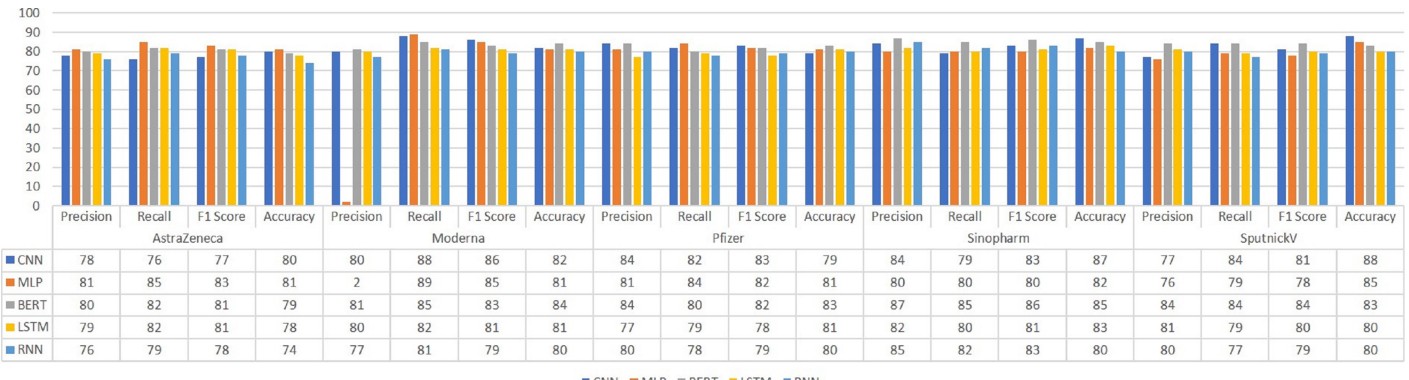

| | Precision | Recall | F1 Score | Accuracy | Precision | Recall | F1 Score | Accuracy | Precision | Recall | F1 Score | Accuracy | Precision | Recall | F1 Score | Accuracy | Precision | Recall | F1 Score | Accuracy |
|---|---|---|---|---|---|---|---|---|---|---|---|---|---|---|---|---|---|---|---|---|
| | | AstraZeneca | | | | Moderna | | | | Pfizer | | | | Sinopharm | | | | SputnickV | | |
| CNN | 78 | 76 | 77 | 80 | 80 | 88 | 86 | 82 | 84 | 82 | 83 | 79 | 84 | 79 | 83 | 87 | 77 | 84 | 81 | 88 |
| MLP | 81 | 85 | 83 | 81 | 2 | 89 | 85 | 81 | 81 | 84 | 82 | 81 | 80 | 80 | 80 | 82 | 76 | 79 | 78 | 85 |
| BERT | 80 | 82 | 81 | 79 | 81 | 85 | 83 | 84 | 84 | 80 | 82 | 83 | 87 | 85 | 86 | 85 | 84 | 84 | 84 | 83 |
| LSTM | 79 | 82 | 81 | 78 | 80 | 82 | 81 | 81 | 77 | 79 | 78 | 81 | 82 | 80 | 81 | 83 | 81 | 79 | 80 | 80 |
| RNN | 76 | 79 | 78 | 74 | 77 | 81 | 79 | 80 | 80 | 78 | 79 | 80 | 85 | 82 | 83 | 80 | 80 | 77 | 79 | 80 |

CNN ■ MLP ■ BERT ■ LSTM ■ RNN

**Figure 4 Comparison of deep learning models using Word2Vec features.**

dataset when trained using BoW features. The highest accuracy score is 92% which is achieved by ETC using BoW features. Classifiers using Word2Vec have achieved low results as compared to BoW and feature unions. Besides the highest accuracy, overall performance using BoW features is better. Despite the simple occurrence count, BoW features often show better results as compared to other complex feature engineering approaches.

Figure 4 illustrates the precision, recall, F1-score, and accuracy results comparison of the deep learning models. The highest precision score has been achieved by RNN on the Sinopharm sub-dataset with an 85% score. SGD has achieved the highest precision using BoW and feature union on the Moderna dataset. ETC has achieved the highest precision score using BoW on Sinopharm and SputnickV datasets. BERT has achieved the highest recall on the Moderna dataset and the highest F1-score on the Sinopharm dataset. Deep learning models MLP and BERT have achieved highest 85% accuracy score which is lower than the results achieved with ETC and BoW.

From the above discussion, it is clear that the classifiers show the highest result regarding the sentiment classification of vaccine-related tweets when trained using the BoW features. Overall, ETC has shown the highest results on all datasets. Randomization and optimization features make ETC more efficient in text classification by reducing bias and variance. TF-IDF considers the importance of words and assigns weights accordingly while BoW is a simple and flexible technique and only considers the frequency of unique terms. The feature union of both techniques contains redundant features, also increases the training time, and did not improve the performance of the models. Deep learning models often provide high accuracy on large-sized datasets but with more training time. But in the case of sentiment analysis of COVID-19-related tasks, the deep learning model did not achieve robust results. ETC in combination with BoW is the most suitable approach for the sentiment analysis of COVID-19 vaccine-related tweets.

In this study, the average accuracy score on all datasets is 0.9111 which is comparable to the scores of other studies such as 0.8177 using NB by *Villavicencio et al. (2021)*, 0.9059 using LSTM and 0.9083 by BiLSTM by *Alam et al. (2021)*. However, this study uses 20,967

tweets that are larger than those used in other studies like 993 tweets by *Villavicencio et al. (2021)*. *Alam et al. (2021)* used the same dataset as ours but employed complex deep-learning models and achieved lower results. All these things proved the superiority of the proposed approach.

Deep learning models do not perform well compared to machine learning models used in this study. First of all, the size of the dataset is small and not enough for deep learning models to get a good fit. Second, the data is sparse which leads to poor performance of deep learning models; that is the reason word2vec features also do not perform well using machine learning models. Third, deep learning models need to tune a large number of hyperparameters which require a large-sized dataset. For the current study, the small-sized dataset is not enough to produce good results using deep learning models.

This research has some limitations like the tweets utilized in this study represented just 1% of daily tweets, therefore they might not be an accurate representation of all tweets. In addition, model fine-tuning was restricted to some parameters only due to a lack of resources needed for training; additional parameters were not tweaked. If the tuning had been done more extensively, the performance of these models may have been enhanced much further.

## Performance of manually vs TexBlob annotated dataset

For analyzing the impact of data annotation from different techniques including TextBlob and manual annotation, experiments are performed with both datasets separately. Table 11 shows the accuracy of all the models using both datasets. The results indicate that the performance of the models is marginally better when TexBlob annotated dataset is used. Although manual annotation is considered the best for machine learning models, their performance is better using the TextBlob dataset. Since the models and TextBlob follow a similar mechanism of objectivity, it is possible that TextBlob makes mistakes similar to machine learning models. Also, since the machine learning models work on features, TextBlob may be providing more correlated features to the models for training which increases their performance. The performance using VADER annotation is inferior to TextBlob.

## Trend analysis and future directions

The distribution of stances into two categories: 'Against' and 'In favor' is presented in Fig. 5 which considers the tweets according to vaccine types. It can be observed that the 'Against' is the dominant stance found in the total tweets. On the entire dataset, the number of tweets for the 'Against' stance is 60% while only 40% belongs to the 'In favor' stance. It also shows that the tweets related to Moderna and SputnikV vaccines have the highest number of 'against' stances as compared to other vaccines.

The results suggest that the trends against and in favor of vaccines have similar patterns indicating that the majority of tweets contain negative sentiments regarding the vaccination process. The highest number of tweets 'In favor' is for 'Sinopharm' where 46% favor this vaccine. Similarly, the SputnikV vaccine has the highest number of 'Against' tweets, which are 65%.

**Table 11 Accuracy of models with TextBlob and manually labeled dataset using BoW features.**

| Dataset | Model | TextBlob | VADER | Manual annotation |
|---|---|---|---|---|
| AstraZeneca | RF | 87% | 85% | 85% |
| | ETC | 89% | 87% | 86% |
| | GBM | 86% | 85% | 82% |
| | LR | 87% | 87% | 83% |
| | NB | 81% | 79% | 78% |
| | SGD | 88% | 85% | 85% |
| Moderna | RF | 90% | 87% | 88% |
| | ETC | 91% | 91% | 90% |
| | GBM | 84% | 82% | 84% |
| | LR | 91% | 89% | 89% |
| | NB | 85% | 82% | 81% |
| | SGD | 92% | 88% | 90% |
| Pfizer | RF | 85% | 86% | 88% |
| | ETC | 90% | 88% | 90% |
| | GBM | 85% | 85% | 86% |
| | LR | 88% | 87% | 88% |
| | NB | 85% | 86% | 83% |
| | SGD | 89% | 89% | 89% |
| Sinopharm | RF | 90% | 88% | 90% |
| | ETC | 92% | 91% | 92% |
| | GBM | 90% | 87% | 90% |
| | LR | 89% | 86% | 89% |
| | NB | 81% | 78% | 81% |
| | SGD | 91% | 88% | 91% |
| SputnikV | RF | 92% | 89% | 90% |
| | ETC | 92% | 91% | 90% |
| | GBM | 86% | 86% | 87% |
| | LR | 90% | 85% | 88% |
| | NB | 80% | 77% | 81% |
| | SGD | 91% | 91% | 89% |

This study also performs an analysis of people's opinions with time. People's opinion changes with time. Figure 6 presents how sentiments of tweets vary with time. In favor sentiments are in green color and Against sentiments are in red. Fluctuation or Variation of sentiments can be seen clearly in Fig. 6. The 'In favor' sentiments are at their peak on March 21 at the end of the final trial of vaccines, while the 'Against' sentiments are at a high rate in the mid of April 21 and May 21.

Furthermore, we have analyzed keywords and themes in each sentiment class that are in favor and against. Table 12 presents that 'In Favor' of vaccine tweets are related to hope, support and happiness. On the other hand, 'Against' is related to fear, anger, and disappointment.

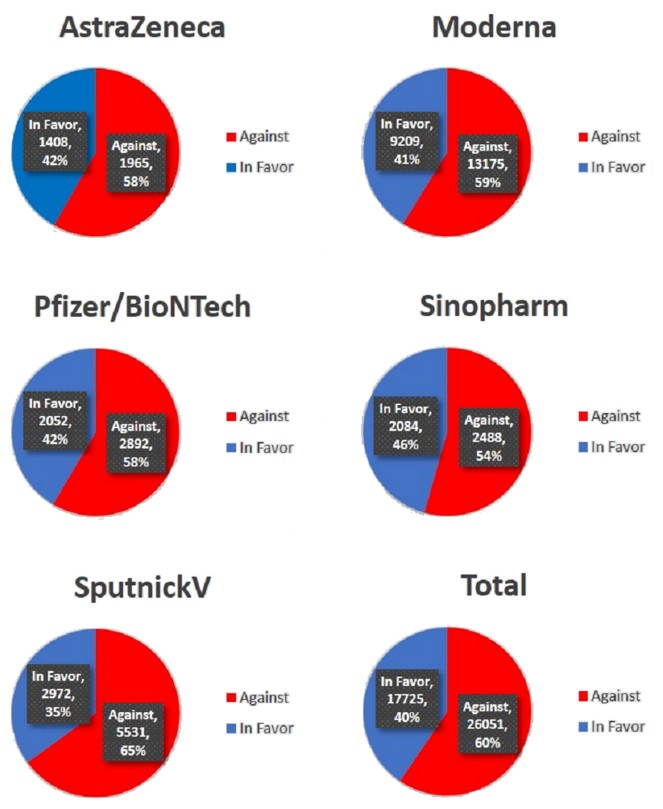

**Figure 5 Comparison of sub-datasets regarding sentiment trends.**

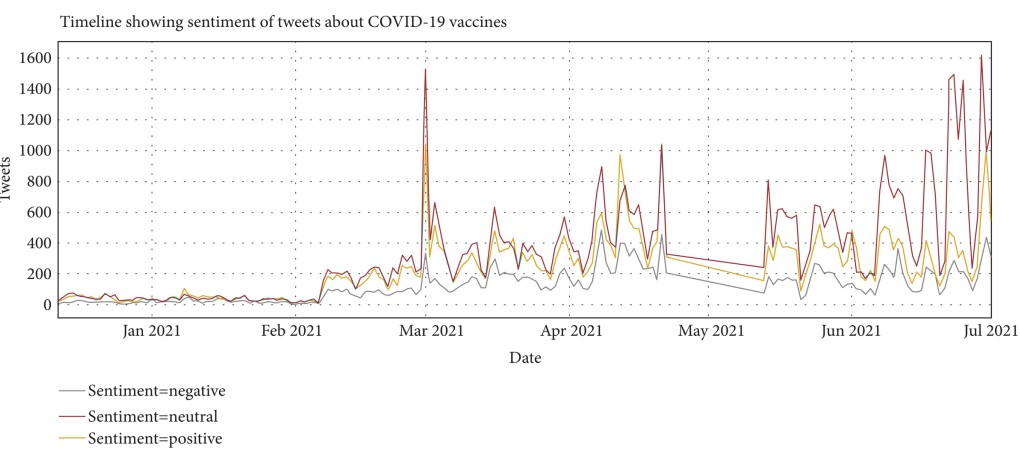

**Figure 6 Variation of sentiments over time.**

Potential future directions of this work could be to analyze public trends regarding more vaccine types by employing natural language processing techniques. Moreover, government and other relevant agencies should provide more detail about the effectiveness and advantages of vaccines to increase the trust of the general public. Public trends for vaccines regarding gender and age groups can also be analyzed in the future. Twitter has been a source for a large number of research studies for sentiment analysis. However, the

**Table 12 Keywords and themes in favor and against polarities.**

| Sentiment polarity | Theme | Keywords |
|---|---|---|
| In favor | Hope | Innovators, fitness, favouring, fortunes, adventures, confident, overcoming, efficiency, winnings, productivity, fascinating |
| | Happiness | Congratulates, thrill, appeal, cheerful, motivator |
| | Support | Aid, kindness, pardoned, wished, truthfulness, greatness, devoting, appreciating, consulted, facilitate, propel, assistance |
| Against | Fear | Adverse, torturous, hoard, wrecks, poisoned |
| | Anger | Disagreeing, shooting, angering, aggravated, outrage, terrifies, crazies, hates |
| | Disappointment | Stealing, misguide, misleads, fool, scammed, impossible, discouraged, blackmail, poisoned, pocketed, disregarding, slanders |

**Table 13 The acronyms used in this manuscript.**

| Acronyms | Definition |
|---|---|
| ANN | Artificial neural network |
| BERT | Bidirectional encoder representations from transformers |
| BoW | Bag of words |
| CDC | Centre of disease control |
| CNN | Convolutional neural network |
| COVID-19 | Coronavirus disease of 2019 |
| ETC | Extra tree classifier |
| GBM | Gradient boosting machine |
| IoT | Internet of things |
| LDA | Latent Dirichlet allocation |
| LR | logistic regression |
| LSTM | long short-term memory |
| MLP | Multilayer perceptron |
| NB | Naive Bayes |
| RF | Random forest |
| RNN | Recurrent neural network |
| SGD | Stochastic gradient descent |
| TF-IDF | Term frequency-inverse document frequency |
| UN | United nations |
| WHO | World health organization |

probability of fake and biased tweets can not be ignored. Several studies point out a higher ratio of false/biased news as high as 25%, especially for the political campaigns (*Bovet & Makse, 2019*; *Vosoughi, Roy & Aral, 2018*; *Shao et al., 2018*). However, a recent study reveals that the ratio of biased tweets or misinformation related to COVID-19 is approximately 3.29% (*Sharma et al., 2020*). Considering this ratio of biased tweets, for the

current study we do not handle the biased tweets aspect and intend to incorporate it in the future study. The acronyms used in the manuscript are presented in Table 13.

This study has several limitations. Since it is based on the manual feature engineering approach BoW in combination with the machine learning model and has inherent limitations of BoW. BoW neglects word semantics and the use of grammar. It simply provides the terms' occurrence and ignores the semantic importance of terms. In the future, we will explore more feature engineering techniques to improve the sentiment analysis task.

## CONCLUSION

This research explores the opinion dynamics related to the COVID-19 vaccine by performing sentiment analysis on different vaccine-related tweets. The dataset is divided into five subsets and investigated separately to get deep insights and quantitative assessment. For sentiment analysis, several machine learning models coupled with four feature representation techniques (TF-IDF, BoW, Word2Vec, and feature union) have been compared. The results show that the ETC with BoW has the highest accuracy of 92% for sentiment analysis of COVID-19-related tweets. Predominantly, the performance of models is better when used with the BoW features. Deep learning models tend to show poor performance with the current dataset, as compared to machine learning models. Empirical and trend analysis of COVID-19 vaccine-related tweets reveals that the spread of unreliable and misinformation is increasing on social media platforms. Deep insights show that 60% are against vaccines. Furthermore, it was observed that a large number of 'Against' are for the SputnikV vaccine, followed by Moderna. Temporal analysis indicates that the ratio of 'In favor' sentiments for COVID-19 vaccination has been elevated over time. For future work, the analysis regarding age groups and gender can be incorporated for vaccination-related trend analysis.

### Funding
The authors received no funding for this work.

### Competing Interests
Imran Ashraf is an Academic Editor for PeerJ.

### Author Contributions
- Shoaib Ahmed conceived and designed the experiments, analyzed the data, prepared figures and/or tables, and approved the final draft.
- Dost Muhammad Khan conceived and designed the experiments, analyzed the data, prepared figures and/or tables, and approved the final draft.
- Saima Sadiq conceived and designed the experiments, analyzed the data, prepared figures and/or tables, and approved the final draft.
- Muhammad Umer conceived and designed the experiments, analyzed the data, prepared figures and/or tables, and approved the final draft.

- Faisal Shahzad performed the experiments, performed the computation work, authored or reviewed drafts of the article, and approved the final draft.
- Khalid Mahmood performed the experiments, performed the computation work, authored or reviewed drafts of the article, and approved the final draft.
- Heba Mohsen performed the experiments, performed the computation work, authored or reviewed drafts of the article, and approved the final draft.
- Imran Ashraf performed the experiments, performed the computation work, authored or reviewed drafts of the article, and approved the final draft.

## Data Availability

The code and data is available at GitHub: https://github.com/MUmerSabir/CovidVaccine; MUmerSabir. (2022). MUmerSabir/CovidVaccine: PeerJ Covid Vaccine (PeerJVaccine). Zenodo. DOI 10.5281/zenodo.7298734.

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
