# Peer review of "Temporal analysis and opinion dynamics of COVID-19 vaccination tweets using diverse feature engineering techniques"

_PeerJ Computer Science, doi:10.7717/peerj-cs.1190_

## Round 0.1 · original submission · Major Revisions

Dear Authors,

Reviewers are suggesting some improvements especially related to adding more details in the Introduction and experiment parts. The novelty of the work also needs to be highlighted.

·

Basic reporting

There are several concerns with this manuscript in its current form.
 
This paper proposes a ML framework for sentiment analysis based on Tweets which are related to COVID-19 vaccines. Additionally, different feature engineering approaches were tested toward providing the suitable one in terms of accuracy.
 
The abstract should be revised and connected to the introduction. When the title is read as the introduction, they are related to the same purpose, which is: to explore feature engineering techniques considering sentiment analysis regarding COVID19 vaccination. However, it does not happen in the abstract, it is not clear what is the real meaning of the manuscript. The abstract should be revised as it does not enough chiefly introduce the area of research along with the research question.

Experimental design

The introductory part should provide more information and contributions to the proposal instead of defining theoretical issues. 
 
Acronym table needed to be added at the end of the paper.
 

In LSTM detail, reference is not properly cited. It is showing up (?) on line 293.
 
The python packages used should be included with reference.

Validity of the findings

Main theme of paper is sentiment classification and diverse feature engineering. Authors need to cite some state-of-the-art approaches based on these two topics in related work to make it interesting. How much work is done on these topics and still continue.
 Any reason why deep learning model not performs well? 
Limitations of proposed work needed to be added.

Additional comments

Suggestions:
- In general, the learning models seems little scared. They should be in structured way. 
- The novelty and contributions of your work should be better explored.

Reviewer 2 ·

Basic reporting

The paper presents sentiment analysis of tweets on different COVID-19 vaccinations. The dataset "COVID-19 All Vaccines Tweets" is downloaded from Kaggle. A bunch of classical machine learning models including Random Forest, Naive Bayes, logistic regression, SGD, ETC, ANN, and CNN were analyzed and compared. Overall, the paper is well organized and straightforward to follow. However, considering the dataset adopted and the learning methods evaluated, the paper is more of a fundamental application of machine learning models on a publicly accessible dataset. Then to make something interesting authors manually labelled the dataset which is a good approach. Still for that, I have the following concerns:

Concerns:
(1) No details are found how the authors manually labelled the dataset in the dataset description section.
(2) Based on this paper, "TF", "TF-IDF", and "word-embeddings" are the three data representation schemes. I strongly recommend to add a table showing advantages and disadvantages of the three.
(3) All the machine learning methods introduced in this paper are traditional and fundamental, no need to explain them in detail by adding equations. In order to make paper compact, Add them all in a two-column table with proper referencing.
(4) For the deep learning models presented in this paper, the authors used a 2x2 max-pooling layer in CNN. Since the tweet data is text-based and sequential, one-dimensional convolution is often recommended to process it.
(5) Also, the authors should add the limitations of proposed approach.

Experimental design

See the above section.

Validity of the findings

See the above section

Additional comments

Minor format and writing issues:
The dataset is divided into two classes: ’against’ and ’in favor’. Tweets classified into the former class present positive opinions of users", 'against' and 'in favor' should be switched to make the meanings consistent with the explanation. They should need to strict with one label convention either positive, negative or against, in favor.

---

## Round 0.2 · accepted · Accept

The revised article is ready for acceptance.

·

Basic reporting

Addressed Issues are improved , no comment

Experimental design

no comment

Validity of the findings

no comment

Additional comments

I am pretty much satisfied with the revised version of the paper. The authors carefully address my concerns. I suggest accepting the research work in its current form.

Reviewer 2 ·

Basic reporting

Well, the revised version of the paper looks more suitable for publication than the first one. I think now the paper's quality is good enough to get published. Therefore, my suggestion is to accept the paper.

Experimental design

See above

Validity of the findings

See above